# Operando Synthesis of High-Curvature Copper Thin Films for CO_2_ Electroreduction

**DOI:** 10.3390/ma12040602

**Published:** 2019-02-17

**Authors:** Xin Zhao, Minshu Du, Feng Liu

**Affiliations:** 1School of Materials Science and Chemical Engineering, Xi’an Technological University, Xi’an 710021, China; helenzhaoxin@gmail.com; 2School of Materials Science and Engineering, Northwestern Polytechnical University, Xi’an, Shaanxi 710072, China

**Keywords:** CO_2_ electroreduction, copper, thin films, high-curvature morphology, cathodic potential, CO_2_ flow rate

## Abstract

As the sole metal that could reduce CO_2_ to substantial amounts of hydrocarbons, Cu plays an important role in electrochemical CO_2_ reduction, despite its low energy efficiency. Surface morphology modification is an effective method to improve its reaction activity and selectivity. Different from the pretreated modification method, in which the catalysts self-reconstruction process was ignored, we present operando synthesis by simultaneous electro-dissolution and electro-redeposition of copper during the CO_2_ electroreduction process. Through controlling the cathodic potential and CO_2_ flow rate, various high-curvature morphologies including microclusters, microspheres, nanoneedles, and nanowhiskers have been obtained, for which the real-time activity and product distribution is analyzed. The best CO_2_ electro-reduction activity and favored C_2_H_4_ generation activity, with around 10% faradic efficiency, can be realized through extensively distributed copper nanowhiskers synthesized under 40 mL/min flow rate and −2.1 V potential.

## 1. Introduction

Excessive CO_2_ emission has caused severe climate and environment problems, which is breaking the sustainability of human society. Thus, there is an urgent demand for converting CO_2_ into fuels or commodity chemicals. Compared with other methods of CO_2_ conversion, including thermochemical, photochemical, and biochemical methods, the electrochemical reduction method is particularly important and attractive in view of the following merits: (1) Feasible combination with renewable sources like solar, wind, and nuclear energy; (2) low cost and safe operating conditions; (3) easy control over reaction pathways via changing the potential, electrocatalysts, and electrolytes. In recent years, there has been plenty of interest absorbed into the electrochemical reduction of CO_2_, which reduces carbon dioxide to multi-hydrocarbons at ambient temperature [1,2,3,4], facilitating the carbon cycle and relieving energy problems. Amongst various metal electrocatalysts studied in aqueous systems, Cu has a unique ability to reduce carbon dioxide to substantial amounts of hydrocarbons, such as CH_4_, C_2_H_4_, and HCOOH [5,6,7]. Unfortunately, the energy efficiency of Cu for CO_2_ electroreduction is low, such that high over-potential must be performed to activate inert CO_2_ molecules, which gives rise to many efforts for improving activity and selectivity of the CO_2_ reduction reaction (CO_2_RR) on copper.

The intrinsic complexity of multi-electron transfer reaction, CO_2_RR activity, and selectivity was highly affected by the composition and morphology of catalysts, electrolytes, and pH value. Various metal catalysts could be categorized into four groups based on their product distributions: (1) Cu, the only metal capable of reducing CO_2_ to hydrocarbons at significant rates; (2) Au, Ag, Zn and Pd, the major product of each is CO; (3) Pb, Hg, In, Sn, Cd and Bi, which primarily produce formate; and (4) Ni, Fe, Pt and Ti, where only hydrogen evolution is observed, instead of CO_2_RR activity, at the steady state [8]. Oxide-derived metallic catalysts exhibit superior CO_2_RR performance, as the intentional oxidation and reduction of metallic electrodes could contribute to more active surface sites [9,10,11]. For example, oxide-derived Au has shown the highest CO production faradaic efficiency, 96%, in 0.5 M NaHCO_3_ [10]. Electrolytes were seen to be a key role in controlling the selectivity of the reaction, because of the different nature of the ions in the electrolyte [12]. For instance, cationic species (Li^+^, Na^+^, K^+^ and Cs^+^) in bicarbonate solutions can be used to control the CH_4_/C_2_H_4_ ratio [13]. In addition, pH is also an important parameter for CO_2_RR. CO_2_ reduction is usually carried out in bicarbonate electrolytes at a close-neutral pH, since CO_2_ acts as a buffer. For CO reduction at a pH of 6~12, the CH_4_ formation is pH dependent, while C_2_H_4_ is independent of pH [14]. For selectivity of Cu, dilute KHCO_3_ electrolytes with an alkaline pH results in high selectivity for C_2_H_4_ [15]. When the concentration of bicarbonate is high the local pH remains close to neutral, favoring CH_4_ and H_2_ production [16]. Modifying the surface morphology of copper has been concluded to be an effective method to improve its activity and selectivity toward the CO_2_ electro-reduction [17,18,19,20]. Generally, different pretreatments were used to change morphologies of copper electrodes before the electrochemical measurement of CO_2_RR. For example, the electrodes, which were electropolished, covered with nanoparticles, and sputtered with argon ions, provided an abundance of under-coordinated sites on the roughened surface (nanoparticle covered surface) and enabled higher selectivity towards hydrocarbons [17]. In addition, the bromide-promoted morphology of copper dendrites was proven to be a highly selective electrocatalyst, which reduced CO_2_ to ethylene with a faradic efficiency of 57% at a high current density of 170 mA/cm^2^ [21]. The selectivity of these kinds of copper electrodes was caused by the high-index faces and the under-coordinated sites on the high-curvature structures; the identification of active sites and the recognition of corresponding catalytic mechanisms were generally based on such pretreated morphology. Thus, a key question is whether the pretreated morphology could be held unchangeable during the electrochemical reduction. Actually, most catalysts undergo morphology self-reconstruction during measurements [22,23], which is always neglected. Thus, the true morphology of copper during CO_2_ electroreduction is unknown, which makes it difficult to identify the real catalytically active sites and hinders the understanding of catalytic mechanisms.

Here, operando synthesis of high-curvature morphology of copper thin film was realized by application of simultaneous electro-dissolution and electro-redeposition during electrochemical reduction of CO_2_. Subjected to different bias potentials and different flow rates of CO_2_, a morphological evolution of copper from original granular, to microclusters, microspheres, nanoneedles, and nanowhiskers respectively, was observed. The resultant real-time CO_2_RR activity and product selectivity were also discussed. As a result, the correlation between the property and structure of such kind of copper catalyst is straightforward, which improves the understanding of CO_2_RR mechanism and offers guidelines of rational design for advanced electrocalysts.

## 2. Experimental Section

### 2.1. Preparation of Original Copper Thin Film Electrode. 

Copper thin film was prepared by electrodeposition in an aqueous electrolyte composed of 0.25 M copper sulfate solution and 50 mM sulfuric acid. NiTi shape memory alloy (SMA) was a typical smart metal that could remember its original shape after deformation, when heated, due to its reversible martensitic transformation [24]. A near-equiatomic NiTi sheet, sized of 10 mm × 10 mm × 1 mm, was used as the substrate for electrodepostion due to its negligible CO_2_RR activity and good chemical stability in the electrolyte. In addition, in our later work, the two-way shape memory effect of NiTi [25] was used to induce elastic strain to Cu nanofilm and study the strain effect [26] on the CO_2_ reduction reaction of Cu. A three-electrode cell including NiTi as working electrode, Pt mesh as counter electrode, and Ag/AgCl (1M Cl^−^) as reference electrode was used. All potentials reported in this paper are quoted with Ag/AgCl. The polycrystalline Cu film was deposited at room temperature by chronoamperometry at −0.34 V. After 300 s deposition, the electrode was taken out and rinsed gently with deionized water several times. 

### 2.2. Operando Synthesis of High-Curvature Copper During Electroreduction Of CO_2_

Electroreduction of CO_2_ was conducted at the ambient temperature in an H-type two compartment cell, as shown in Figure 1. A standard three-electrode system was used, employing Cu/NiTi working electrode, a Pt mesh counter electrode, and an Ag/AgCl reference electrode. A CO_2_-saturated 0.1 M KHCO_3_ aqueous solution was used as the electrolyte for CO_2_RR. Firstly, a linear sweep voltammetry (LSV) technique was employed to get the CO_2_ RR activity of the original copper thin film, and then a chronoamperometry technique was used to perform CO_2_RR for 1 h at four specific potentials with continuous CO_2_ bubbles in the solution, during which process the electro-dissolution and electro-redeposition of copper ions occurred and resulted in high-curvature morphology. In addition, the CO_2_ flow rate effect on the surface morphology of operando synthesized copper was studied.

### 2.3. Characterization of CO_2_RR Activity and Products Distribution of High-Curvature Copper

I-t curves at different bias potentials during operando synthesis of copper were shown to evaluate the activity of CO_2_RR, and a gas chromatograph (GC) was connected to the vent of H-type electrochemical cell for gaseous products real-time analysis. The GC was equipped with a thermal conductivity detector (TCD) to quantify hydrogen and a hydrogen flame ionization detector (FID), equipped with a methanizer to quantify carbon monoxide, methane, and ethylene, was connected to the vent of the H-type electrochemical cell for real-time gaseous products analysis. The parameters of the GC were set as follows: Oven temperature was 360 °C, TCD temperature was 120 °C, FID temperature was 150 °C, and the column temperature was 70 °C.

### 2.4. Characterization of Surface Morphology and Composition of Copper Thin Films

Samples were rinsed gently with deionized water after a 1 h reaction. The surface morphology and composition of the Cu film, synthesized in situ under different potentials and different CO_2_ flow rates, were analyzed through a scanning electron microscope (SEM) and energy spectrum analysis (EDS). 

## 3. Results and Discussion

### 3.1. Similar Morphology and Catalytic Activity of Original Electrodeposited Films

Original morphology of electrodeposited copper thin film is shown in Figure 2a, which indicates the typical granular morphology arising from island growth in eletrodeposition. It is noted that four samples swept to different end potentials had a similar granular morphology with the similar roughness and sub-micro particle size. Linear sweep voltammetry (LSV), at a scan rate of 50 mV/s, is used to provide a quantitative assessment of the catalytic performance of four studied electrodes. We note that they have the same onset potential of about −0.9 V and almost the same CO_2_RR kinetics (indicated by the slope of LSV curves in the Figure 2b).

### 3.2. Effect of Cathodic Potential on the Morphology and Product Selectivity of Copper Films

It is known that cathodic potential, as an important parameter in CO_2_RR, not only determines the pathway of reaction but also affects the copper reconstruction process (shown schematically in Figure 3), e.g., more negative potentials could facilitate the nucleation rate of copper [27,28,29,30]. Therefore, it would be informative to probe the correlations between the as-formed morphology and the resultant CO_2_RR activity. Depending on the applied potential, different structural morphologies emerged. Various morphologies of operando-synthesized copper could be observed, shown in Figure 4. At −1.7 V, the surface is mainly covered with microspheres, with a diameter of about 4~6 μm, and scattered nanometer needles, with a length of about 1~3 μm, which changed to a flower-like nanoneedle, with a length of about 5~7 μm, at −1.9 V. As the overpotential increased to −2.1 V, sharper nanowhiskers, with the length of about 6~10 μm and a length-to-diameter ratio of 30~70, appeared rapidly and then turned into a uniformly dispersed short length of about 3 μm nanowhiskers, with a length-to-diameter ratio of 5~10, at −2.3 V. This morphology evolution was due to tip effect and hydrogen bubble generation (which served as dynamic template for redeposition) during the simultaneous dissolution and redeposition of Cu ions. We also analyzed the element composition of the high-curvature morphologies (shown in Figure 5 and Table 1). The high-curvature morphology is composed of a mixture of Cu and its oxidation state. Such copper oxides would contribute to relatively higher CO_2_RR activity than pure Cu [31]. 

CO_2_RR activity of various high-curvature coppers was revealed from chronoamperometric i-t curves. With continuous CO_2_ bubbled at a flow rate of 40 mL/min, the chronoamperometry technique was adopted to perform CO_2_RR for 1 h at four specific potentials. As shown in Figure 6c, the reduction current increased with more negative potentials. Meanwhile, the total reduction current of different reaction potential was stable. Though the total reaction current was as high as about 45 mA/cm^2^ at −2.3 V, it was revealed by gas chromatography that the major reduction current owed to the electroreduction of water (2H_2_O* + 2e^−^→H_2_ + 2OH^−^) instead of CO_2_. For CO_2_RR, the main gas products were CO, CH_4_ and C_2_H_4_, which were monitored every 15 min. As shown in Figure 6a, the percentage of the three main CO_2_RR products changed with bias potentials. At −1.7 V and −1.9 V, CO was the dominant product with percentages of 59% and 73%. When the more negative potential was applied for operando synthesis, like −2.1 V and −2.3 V, C_2_H_4_ proportion increased to 65% and 60% of the three main gas products as a preferred product. It was speculated that higher local electric field around high-curvature nanowhiskers of copper could improve bubble nucleation rates and cation stabilization, which favored C_2_H_4_ formation during electroreduction of CO_2_ [15,27,32]. As was shown by the energy spectrum, high-curvature morphology contained the oxidation state of Cu, which also made the electrode exhibit good ethylene selectivity [33,34]. Additionally, we also compared C_2_/C_1_ product ratios of various high-curvature copper (Figure 6b), and found that widely distributed nanowhiskers of copper had a high C_2_/C_1_ product ratio (about 1:6), which was more than five times than that synthesized under −1.7 V and −1.9 V. Moreover, C_2_/C_1_ product ratio of different morphological copper fluctuated slightly within 1 h of the operando synthesis process, as shown in Figure 6b, and it was surprising to see an increasing C_2_/C_1_ product ratio of nanorod copper, synthesized under −2.3 V, from 1.1- at the first quarter to 1.8 at the fourth quarter of the hour. This be caused by the increase of the nanowhisker distribution area in the copper electro-dissolution and electro-redeposition process during 1 h CO_2_RR. 

### 3.3. Effect of CO_2_ Flow Rate on the Morphology and Product Selectivity of Copper Films

Flow rate was another important parameter that affected the mass transfer of CO_2_, the copper reconstruction process, and CO_2_ electroreduction [33,34,35]. In the following, the effects of CO_2_ flow rate on the morphology of operando synthesis copper and the resulting CO_2_RR activity and selectivity were studied. Here, various morphologies of copper were synthesized in situ under different flow rates of CO_2_, 20 mL/min, 40 mL/min, 60 mL/min, and 80 mL/min, with the same bias potential at −2.1 V for 1 h in the solution. Under the lower flow rate of 20 mL/min, the morphology of copper surface became rougher and only short and thick microclusters (about 1 μm thick) dispersed on the granular copper. Cu nanowhiskers morphology emerged under 40 mL/min flow rate, a combination of micro polygon particles and rod-like copper dispersed sparsely under 60 mL/min flow rate, and a relative flat surface without any high-curvature structures exhibited when CO_2_ was bubbled at a flow rate of 80 mL/min (Figure 7). We also employed the chronoamperometry technique to perform CO_2_RR for 1 h at four different flow rates. As expected, less active sites on the flat surface of copper rendered to the lower activity of CO_2_RR (Figure 8d), and overall current density was about 28 mA/cm^2^ under an 80 mL/min flow rate (compared to about 40 mA/cm^2^ under a lower flow rate). Nevertheless, the total reduction current under each CO_2_ flow rate still remained stable. More distinct structure-property correlation was revealed by faradaic efficiency (FE) of gaseous products of various morphologies of copper, shown in Figure 8a–c. Optimal faradic efficiency for CO_2_RR was obtained on copper nanowhiskers at the third quarter of an hour under 40 mL/min flow rate, which was 1.8% ± 0.3% for CO, 1.6% ± 0.24% for CH_4_, and 9.9% ± 1.3% for C_2_H_4_, respectively, and the total current density was about 43 mA/cm^2^. When comparing the overall faradaic efficiency during an hour of CO_2_ electroreduction, nanowhiskers exhibited higher faradic efficiency for C_2_H_4_ than the other morphologies and microcluster copper synthesized under 20 mL/min flow rate performed at a higher faradic efficiency for CH_4_, about 5%. Combinational morphology of micro polygon and rod-like particles presented the lowest faradic efficiency for CO and CH_4_ production. 

### 3.4. Operando-Synthesized Mechanism and Structure-Property Correlation

As shown above, proper cathodic potential and CO_2_ flow rate should be chosen in order to obtain the proper ratio of nucleation rate to the directional growth rate of copper, which ensures operando synthesis of various high-curvature coppers. Presumably, high-curvature morphologies originated from constricted lateral growth due to hydrogen bubbles and the concentrated electric field around tips for copper redeposition. The observed change in morphology correlated directly with CO_2_RR activity and selectivity, that is, copper nanowhiskers with larger length-to-diameter ratio exhibited higher faradic efficiency and favor ethylene production. As studied previously, sharp nanowhiskers could improve bubble nucleation, concentrate stabilizing cations and exhibit high field locally [27], all of which, along with higher local pH, boost the reaction, limiting the protonation of bound CO* intermediate that leads to ethylene formation. 

## 4. Conclusions

In summary, various high-curvature morphologies of copper were operando synthesized during a CO_2_ electroreduction process by controlling cathodic potential and CO_2_ flow rate. The electrodeposited granular copper evolved to microspheres, nanoneedles, and nanowhiskers, respectively, when biased at −1.7 V~ −2.3 V under 40 mL/min flow rate. When biased at −2.1 V with different flow rates of CO_2_, rough surfaces with scattered microcluster morphology emerged under 20 mL/min flow rate, and extensive nanowhisker morphology emerged under 40 mL/min. Higher flow rates of 60 and 80 mL/min failed to form sharp whisker-like morphology but scattered distributed particle-rod-combination morphology and non-cluster surfaces instead. Benefiting from such in-situ synthesized morphologies, real-time CO_2_RR activity and products selectivity were studied, and the correlation between morphology and activity was also discussed. The reduction product was mainly CO at lower overpotential, while high-valued ethylene became the main reduction product when increasing the cathodic potential and the synthesis of high-curvature morphology. Morphologies synthesized under 20 mL/min and 40 mL/min flow rate showed relatively high CO_2_RR activity, that is, rough surfaces exhibited with microcluster morphology favored methane and ethylene selectivity with C_2_/C_1_ ratio about 1, and exhibited nanowhisker morphology favored ethylene selectivity (around 10% faradaic efficiency) with C_2_/C_1_ ratio of 1.8. According to previous mechanistic studies, the reaction pathways of CH_4_ and C_2_H_4_ differ in the bound CO* intermediates [36,37,38,39]. Both morphologies synthesized under 20 mL/min and 40 mL/min flow rate stabilized the bound CO* and prevented its desorption, which resulted in not only high reaction activity but also the further reaction pathways by hydrogenation of CO* or dimerization of two bound CO*. For rough Cu with microcluster morphology, the two aforementioned pathways were coexisting and equi-favored, thus FE for methane and ethylene were both around 5%. For nanowhisker morphology, CO* hydrogenation to COH* could be suppressed by the existence of copper oxide and high local pH [37,38], thus shifting the reaction towards C_2_H_4_ by stabilizing the OCCOH* intermediates (CO* + CO* + H^+^ + e^−^ → OCCOH* → C_2_H_4_).

Hopefully, such operando synthesized copper and its direct effect of morphology on product distribution could inspire more attention to catalyst self-reconstruction in the design of advanced electrocatalysts and analysis of reaction mechanisms.

## Figures and Tables

**Figure 1 materials-12-00602-f001:**
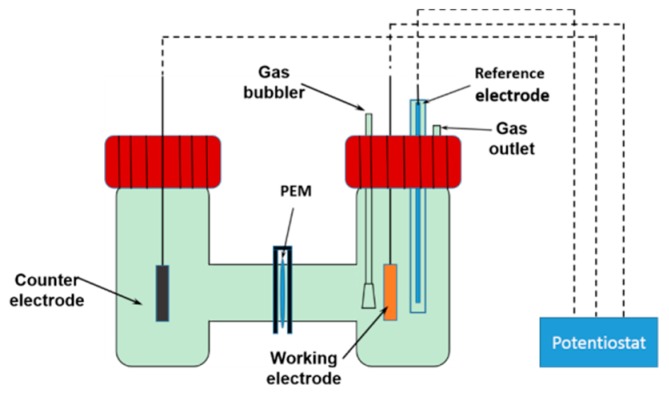
A schematic drawing of H-type cell.

**Figure 2 materials-12-00602-f002:**
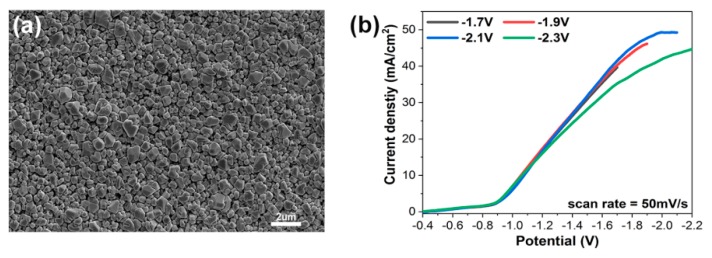
Characterization of electrodeposited copper thin film electrode. (**a**) SEM image of electrodeposited Cu on NiTi substrate in a solutioin of 0.25 M CuSO_4_ and 50 mM H_2_SO_4_. (**b**) Linear sweep voltammogrames of different ending potentials working electrodes in CO_2_-saturated 0.1 M KHCO_3_.

**Figure 3 materials-12-00602-f003:**
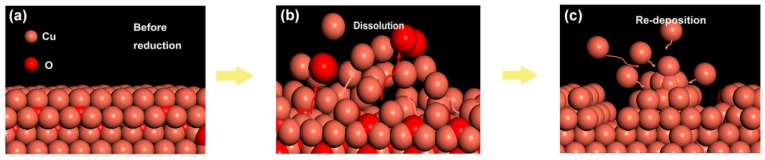
Schematic of reconstruction process, whereby simultaneous dissolution and redeposition occurred. (**a**) Surface morphology of the electrode before the reaction. (**b**) Dissolution of the electrode surface. (**c**) Redeposition of the electrode surface.

**Figure 4 materials-12-00602-f004:**
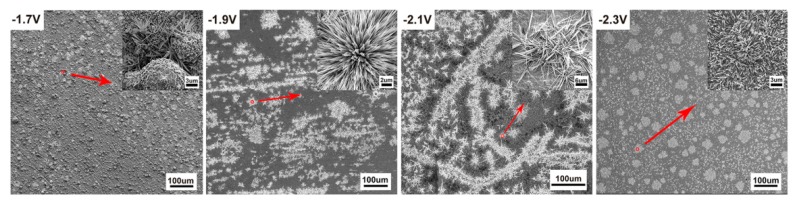
SEM images of different electrodes morphologies synthesized under corresponding bias potentials.

**Figure 5 materials-12-00602-f005:**
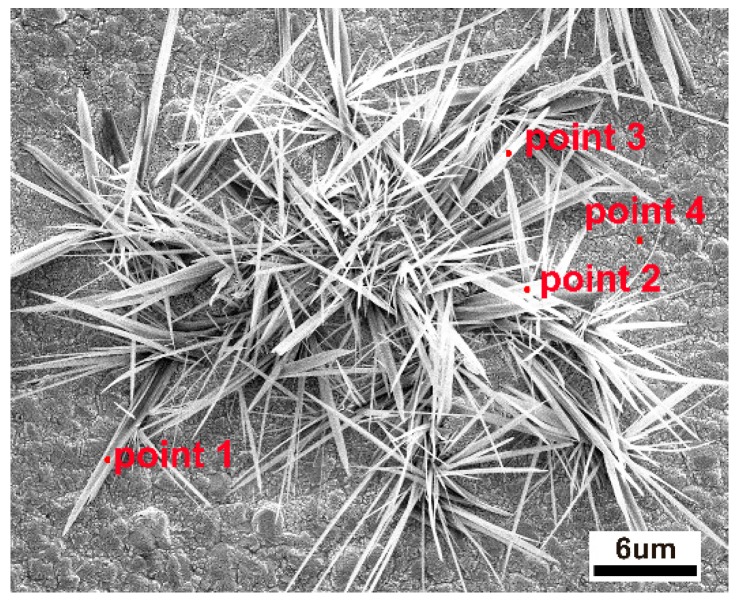
EDS analytical points on the high-curvature morphologies of Cu.

**Figure 6 materials-12-00602-f006:**
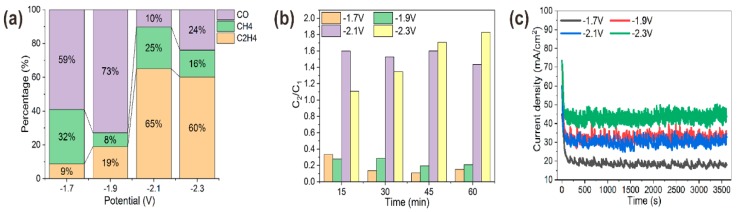
Selectivity characterization of operando synthesized coppers. (**a**) Three main gas products (CO, CH_4_, C_2_H_4_) distribution under different synthesized potential. (**b**) C_2_/C_1_ products ratio of copper synthesized under different potentials for electroreduction of CO_2_, monitored every 15 min for 1 h by gas chromotography. (**c**) Chronoamperometric i-t curves of copper synthesized under different potentials.

**Figure 7 materials-12-00602-f007:**
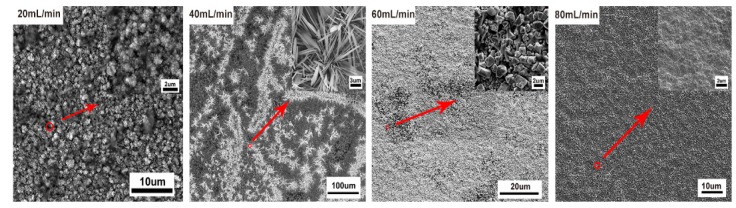
SEM images of copper thin films formed under different flow rates.

**Figure 8 materials-12-00602-f008:**
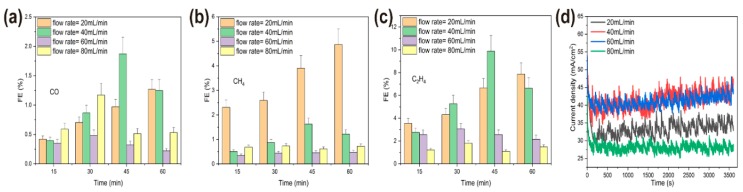
Electrocatalytic CO_2_ reduction performances of operando synthesized coppers under 20, 40, 60, 80 mL/min flow rates. Faradic efficiencies of three main products (**a**) CO, (**b**) CH_4_, (**c**) C_2_H_4_. (**d**) Time-dependent total current density at −2.1 V of different CO_2_ flow rates.

**Table 1 materials-12-00602-t001:** The atomic percentage of copper and oxygen elements on high-curvature morphologies.

	Point 1	Point 2	Point 3	Point 4
Cu (atomic %)	74%	70.5%	71.2%	91.3%
O (atomic %)	26%	29.5%	28.8%	8.7%

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
