# Peer review of "Operando Synthesis of High-Curvature Copper Thin Films for CO2 Electroreduction"

_materials, 2019, doi:10.3390/ma12040602_

Reviewer 1 Report

The work shows an important effect of surface reconstruction of copper catalyst on the CO2 reduction reaction. The manuscript is well written. The content and results are clear with good illustrations and supportive data. Therefore I would like to endorse for publication with a minor check on typo. 

Author Response

Point 1: The work shows an important effect of surface reconstruction of copper catalyst on the CO2 reduction reaction. The manuscript is well written. The content and results are clear with good illustrations and supportive data. Therefore I would like to endorse for publication with a minor check on typo.

Response 1: Thanks for the nice summary of our work and the valuable comments. We have double-check on typo, please see the revised manuscript.

Reviewer 2 Report

The authors present a well-prepared manuscript in my opinion and it should be published in Materials after a slight modification; please add a short paragraph explaining the full experimental details of the GC setup (oven temperature, detector temperature, column etc) and generally please elaborate on the experimental setup procedures more.

Author Response

Point 1: The authors present a well-prepared manuscript in my opinion and it should be published in Materials after a slight modification; please add a short paragraph explaining the full experimental details of the GC setup (oven temperature, detector temperature, column etc) and generally please elaborate on the experimental setup procedures more.

Response 1: We appreciate this positive comment on our work. And as requested, we have added more details about parameters of the GC setup. As indicated in the paragraph, “GC was equipped with a thermal conductivity detector to quantify hydrogen and a hydrogen flame ionization detector equipped with a methanizer to quantify carbon monoxide, methane and ethylene, which was connected to the vent of H-type electrochemical cell for gaseous products real-time analysis. The parameters of GC were set as follow: oven temperature is 360 ℃, TCD detector temperature is 120 ℃, FID detector temperature is 150 ℃, column temperature is 70 ℃.”

Reviewer 3 Report

Authors reported very interesting results from the electrochemical CO2 reduction using the morphological modification of copper electrodes. More quantitative analysis is required for the morphologies they found. Additional references for the comparison with others' CO2 reduction catalysts could be helpful to evaluate their performances .Comments and questions are the following. 

1. Such a schematic drawing would be very helpful for H-type cell.

2. Any reference or more introduction is recommended for the NiTi shape memory alloy and morphology evolution.

3. In Figure 4b, the CO2RR activity was measured up to 1 h. A longer experiment is strongly suggested because the C2/C1 product ratio is increased with time at -2.3 V. A longer time than 1 h has to be tried to see how much the ratio goes up. It is also important as the C-C coupling from C1 to C2 seems to be enhanced only at -2.3 V. Further discussion also needs to be followed on the data.

4. What about C2H6 formation? Did they measure ethane also?

5. Authors distinguished only the particle shapes of the electrodeposited copper thin films obtained with different potentials, but further analysis such as the particle size distributions of microspheres or the length-to-diameter ratio distributions of nano needles or nano whiskers are missing. How much big are the microspheres at -1.7 V? How much shorter are the nanowhiskers at -2.3 V than -2.1 V? More quantitative analysis is required for each SEM image in Figure 3.

6. Why is the optical faraday efficiency (FE) measured at the 3rd quarter in Figure 6? What about a lower flow rate like 20 ml/min?

Author Response

Point 1: Authors reported very interesting results from the electrochemical CO2 reduction using the morphological modification of copper electrodes. More quantitative analysis is required for the morphologies they found. Additional references for the comparison with others' CO2 reduction catalysts could be helpful to evaluate their performances .Comments and questions are the following.

Response 1: Thanks for the valuable comments and advice. Please review our replies to each question as follows:

Point 2:Such a schematic drawing would be very helpful for H-type cell.

Response 2: Thanks for your nice advice and we have added a schematic drawing of H-type cell as Figure 1 in the revised manuscript.

Point 3: Any reference or more introduction is recommended for the NiTi shape memory alloy and morphology evolution.

Response 3: Thanks for the suggestion. We have added several references and more introductions for NiTi in the revised manuscript. As indicated in the paragraph, “NiTi shape memory alloy (SMA) was a typical smart metal that could remember its original shape after deformation when heated due to its reversible martensitic transformation [27, 28]. Near-equiatomic NiTi sheet in size of 10 mm x 10 mm x 1 mm was used as the substrate for electrodepostion due to its negligible CO2RR activity and good chemical stability in the electrolyte. In addition, in our later work, two-way shape memory effect of NiTi [29] was used to induce elastic strain to Cu nanofilm and study the strain effect [30] on the CO2 reduction reaction of Cu.” And the morphology evolution of nanofilm during CO2 electroreduction has seldom been reported before, and the morphology of NiTi substrate should keep being unchanged during the electrochemical measurement since its high chemical stability and no CO2RR activity. As a result, we don't provide any reference about morphology evolution in the manuscript.

Point 4: In Figure 4b, the CO2RR activity was measured up to 1 h. A longer experiment is strongly suggested because the C2/C1 product ratio is increased with time at -2.3 V. A longer time than 1 h has to be tried to see how much the ratio goes up. It is also important as the C-C coupling from C1 to C2 seems to be enhanced only at -2.3 V. Further discussion also needs to be followed on the data.

Response 4: Thanks for pointing out this issue. We have made a longer time CO2RR measurement at -2.3 V, as shown in the figure R1 in the attached PDF. It can be clearly seen that Cu electrode exhibited preferred selectivity for C2 product formation during the first three hours with C2/C1 ratio larger than 1. It semms that C2/C1 ratio increased with time only within the first hour, after reaches to the maximum of 1.6, it fluctuates a little. When reacted 2.5 h, C2/C1 ratio drops gradually with time and falls to be only 0.4 at 5 h. We speculated that C2/C1 ratio increase during the first hour was caused by the increase of nanowhiskers distribution area, and C2/C1 ratio drop may due to the active site occupation by the reaction intermidiates or impurities in the electrolyte that led to lower C-C coupling for C2 product genernation.

Point 5 : What about C2H6 formation? Did they measure ethane also?

Response 5: We indeed measure C2H6 formation, however, only trace amount (below 0.2 ppm) was detected by GC. So we do not show and discuss C2H6 formation in our manuscript.

Point 6 : Authors distinguished only the particle shapes of the electrodeposited copper thin films obtained with different potentials, but further analysis such as the particle size distributions of microspheres or the length-to-diameter ratio distributions of nano needles or nano whiskers are missing. How much big are the microspheres at -1.7 V? How much shorter are the nanowhiskers at -2.3 V than -2.1 V? More quantitative analysis is required for each SEM image in Figure 3.

Response 6: Thank you for your suggestion. We have added quantitative analysis about the size for each SEM image. As indicated in the paragraph, “At -1.7 V, the surface is mainly covered with microspheres with a diameter of about 4 ~ 6 μm, and scattered nanometer needles with a length of about 1 ~ 3 μm, which changed to a flower-like nanoneedle at -1.9V, and the length is about 5 ~ 7 μm. As the overpotential increased to -2.1 V, sharper nanowhiskers appeared rapidly, with the length of about 6 ~ 10 μm and a length-to-diameter ratio of 30 ~ 70 , then turned into a uniformly dispersed short length of about 3 um nanowhiskers at -2.3 V, with a length-to-diameter ratio of 5 ~ 10.”

Point 7 : Why is the optical faraday efficiency (FE) measured at the 3rd quarter in Figure 6? What about a lower flow rate like 20 ml/min?

Response 7: For the optimal FE obtained at the 3rd quarter in Figure 6, we consider it was caused by the largest distribution area of high-curvature nanowhiskers that contributed to more active sites for CO2RR, as the nanowhikers generated by electro-deposition and electro-redeposition of Cu which needs certain time. And in the fourth quarter, decreasing FE may due to the active site occupation by the reaction intermidiates or impurities in the electrolyte.  

As requested, we added data of CO2RR under 20 mL/min flow rate, as shown in Figure 7, only short and thick micro-clusters (about 1 μm thick) dispersed on the granular copper. And its activity and selectivity were indicated in Figure 8 as the orange bars, which showed an increasing FE with time during 1 hour measurement. Similarly, such increasing FE may be caused by more micro-clusters distribution with time. Micro-clusters morphplogy copper had the highest CH4 FE of about 5% compared to other morphologies synthesized under larger flow rate, and relative lower CO FE (about 1.2%) and C2H4 FE (about 8%) compared to nanowhiskers synthesized under 40 mL/min.

Reviewer 4 Report

The present report demonstrates Operando synthesis of various morphological Cu for electroreduction of CO2. Different morphologies were achieved at different cathodic potential and CO2 flow rate. The optimum condition for CO2 electroreduction was 40 ml/min flow rate and -2.1 V applied potential on in situ formed nanowhisker morphologies. Manuscript portrays some important findings which might be useful to the wide scientific community. The “CO2 electroreduction” to hydrocarbons is an extremely important topic envisaged a key to solving future problem of the energy crisis, energy storage, and global warming. I recommend publication of manuscript after few Major revisions.

1. Introduction part is weak and needs to improve by adding more detailed discussion on the importance of CO2 electroreduction, previous works, highest achieved faradaic efficiencies, the effect of morphologies, effect of the composition of materials, pH and electrolyte should be added to increase readability.

2. There is a strong possibility of copper oxides formation which is also equally or more active in CO2 electroreduction. (Science, 2018, 360, 783-787) The chemical composition of nanostructures should be determined using XPS and SEM or TEM elemental mapping/EDX.

3. CO2 may also reduce to liquid products like formic acid, methanol, oxalic acid etc. Did the author analyze liquid products as well? It is recommended to analyze liquid products using GLC and HPLC.

4. There are many grammatical and sentence errors need to be fixed i.e. CO2 “2” should be subscript, NaHCO3 “3” should be subscript, “C2H4” subscript, “ml” should be “mL” in all figure and text, “ration” should be “ratio”, Fig. 6 legends (e) should be “d” In abstract sentence “Different from pretreated ….” Remove “here” In the sentence “Unfortunately, the energy efficiency of…” “performed” should be “applied”, hour should be written “h” reference 23 “Acta” is “Electrochimica Acta” us abbreviation.

5. Why nanowhisker morphology promote better CO2 reduction to ethylene? Any explanation. If appropriate please draw a schematics of the mechanism of CO2 reduction on Cu surface.

Author Response

Point 1: The present report demonstrates Operando synthesis of various morphological Cu for electroreduction of CO2. Different morphologies were achieved at different cathodic potential and CO2 flow rate. The optimum condition for CO2 electroreduction was 40 ml/min flow rate and -2.1 V applied potential on in situ formed nanowhisker morphologies. Manuscript portrays some important findings which might be useful to the wide scientific community. The “CO2 electroreduction” to hydrocarbons is an extremely important topic envisaged a key to solving future problem of the energy crisis, energy storage, and global warming. I recommend publication of manuscript after few Major revisions.

Response 1: We appreciate this positive comment on our work and also thanks for the valuable comments and suggestions. Please review our replies to each question as follows:

Point 2: Introduction part is weak and needs to improve by adding more detailed discussion on the importance of CO2 electroreduction, previous works, highest achieved faradaic efficiencies, the effect of morphologies, effect of the composition of materials, pH and electrolyte should be added to increase readability.

Response 2: Thanks for you suggestion. Following the suggestion, we have added the introduction in the revised manuscript, including discussing the importance of CO2 electroreduction, the effect of morphologies, effect of the compositon of materials, pH and electrolyte. As indicated in the paragraph 1 and 2:

Excessive CO2 emission has caused severe climate and environment problems, which is breaking the sustainability of human society. Thus there is an urgent demand for converting CO2 into fuels or commodity chemicals. Comparing with other methods of CO2 conversion including thermochemical, photochemical and biochemical methods, electrochemical reduction method is particularly important and attractive in the view of following merits: 1) feasible combination with renewable sources like solar, wind and nuclear energy; 2) low cost and safe operating condition; 3) easy control over reaction pathways via changing the potential, electrocatalysts and electrolyte. In recent years, there has been absorbed plenty of interests into the electrochemical reduction of CO2 that reduces carbon dioxide to multi-hydrocarbons at ambient temperature [1-4], facilitating carbon cycle and relieving energy problems. Amongst various metal electrocatalysts studied in aqueous systems, Cu has a unique ability to reduce carbon dioxide to substantial amounts of hydrocarbons, such as CH4, C2H4, HCOOH [5-7]. Unfortunately, the energy efficiency of Cu for CO2 electro-reduction is low, so that high over-potential must be performed to activate inert CO2 molecules, which gives rise to many efforts for improving activity and selectivity of CO2 reduction reaction (CO2RR) on copper.

As the intrinsic complexity of multi-electron transfer reaction, CO2RR activity and selectivity were highly affected by the composition and morphology of catalysts, electrolyte, and pH value.  Various metal catalysts could be categorized into four group based on their products distributions: 1) Cu, the only metal capable of reducing CO2 to hydrocarbons at significant rates; 2) Au, Ag, Zn and Pd, the major product of which is CO; 3) Pb, Hg, In, Sn, Cd and Bi, primarily producing formate; 4) Ni, Fe, Pt and Ti, only hydrogen evolution was observed instead of CO2RR activity at the steady state [8]. And oxide-derived metallic catalysts exhibit superior CO2RR performance, since the intentional oxidation and reduction of metallic electrodes could contribute to more active surface sites [9-11]. For example, oxide-derived Au has shown the highest CO production faradaic efficiency of 96% in 0.5 M NaHCO3 [10]. Electrolyte was seen to be a key role in controlling the selectivity of the reaction, as the different nature of the ions in the electrolyte [12]. For instance, cationic species (Li+, Na+, K+ and Cs+) in bicarbonate solutions can be used to control the CH4/C2H4 ratio [13]. In addition, pH is also an important parameter for CO2RR. CO2 reduction is usually carried out in bicarbonate electrolyte at a close-neutral pH, since CO2 acts as a buffer. For CO reduction at a pH of 6 ~ 12, the CH4 formation is pH dependent, while C2H4 is independent of pH [14]. For selectivity of Cu, dilute KHCO3 electrolytes with an alkaline pH result in high selectivity for C2H4 [15]; while when the concentration of bicarbonate is high, the local pH remains close to neutral, favouring CH4 and H2 production [16].

Point 3: There is a strong possibility of copper oxides formation which is also equally or more active in CO2 electroreduction. (Science, 2018, 360, 783-787) The chemical composition of nanostructures should be determined using XPS and SEM or TEM elemental mapping/EDX.

Response 3: Thank you for pointing out this issue. We have analyzed the element composition of the high curvature morphologies using EDX in SEM. As indicated in the table 1, there is indeed a little copper oxides formation with O atomic percent of 8.7% ~ 29.5% in different sites.

Point 4: CO2 may also reduce to liquid products like formic acid, methanol, oxalic acid etc. Did the author analyze liquid products as well? It is recommended to analyze liquid products using GC and HPLC.

Response 4: Thank you for the suggestion. We indeed tested the liquid products using HPLC, and found that no liquid phase products other than trace amount of formate and thanol were detected, so we only discussed morphology effect on the distribution of gaseous products.

Point 5: There are many grammatical and sentence errors need to be fixed i.e. CO2 “2” should be subscript, NaHCO3 “3” should be subscript, “C2H4” subscript, “ml” should be “mL” in all figure and text, “ration” should be “ratio”, Fig. 6 legends (e) should be “d” In abstract sentence “Different from pretreated ….” Remove “here” In the sentence “Unfortunately, the energy efficiency of…” “performed” should be “applied”, hour should be written “h” reference 23 “Acta” is “Electrochimica Acta” us abbreviation.

Response 5: Thank you for your reminding and we have double-checked the manuscript and made corrections in the revised manuscript.

Point 6: Why nanowhisker morphology promote better CO2 reduction to ethylene? Any explanation. If appropriate please draw a schematics of the mechanism of CO2 reduction on Cu surface.

Response 6: According to previous mechanism studies, the reaction pathways of CH4 and C2H4 differ at the bound CO* intermediates [1-4]. Hydrogenation of the bound intermediate CO* to form COH* results in the formation of methane, while dimerization of the two bound CO* intermediates results in the formation of ethylene. From the thermodynamic point of view, the high curvature morphology consists of copper and its oxidation state, and the interface between Cu+ and Cu0 on the surface stabilizes the CO-CO dimerization and hinders the formation of the C1 pathway[5].

At the same time, experimentally, we also observed that methane production was inhibited, indicating that methane inhibition cannot be explained by thermodynamic effects alone. High curvature morphologies induce high local pH, which is kinetically unfavorable for CO hydrogenation to COH* process, inhibit methane formation, and have a good selectivity twoard ethylene[6-7].

Reference

1.       Schouten, K. J. P., Kwon, Y., van der Ham, C. J. M., Qin, Z.; Koper, M. T. M. A new mechanism for the selectivity to C1 and C2 species in the electrochemical reduction of carbon dioxide on copper electrodes. Chem. Sci. 2011, 2, 1902.

2.       Nie, X., Esopi, M. R., Janik, M. J.; Asthagiri, A. Selectivity of CO2 reduction on copper electrodes: the role of the kinetics of elementary steps. Angew. Chem. Int. Ed. 2013, 52, 2459–2462.

3.       Xiao, H., Cheng, T. & Goddard, W. A. Atomistic mechanisms underlying selectivities in C1 and C2 products from electrochemical reduction of CO on Cu(111). J. Am. Chem. Soc. 2017, 139, 130–136.

4.       Cheng, T., Xiao, H. & Goddard, W. A. Full atomistic reaction mechanism with kinetics for CO reduction on Cu(100) from ab initio molecular dynamics free-energy calculations at 298 K. Proc. Natl Acad. Sci.USA. 2017, 114, 1795–1800.

5.       Xiao, H., Goddard, W. A., Cheng, T. & Liu, Y. Cu metal embedded in oxidized matrix catalyst to promote CO2 activation and CO dimerization for electrochemical reduction of CO2. Proc. Natl Acad. Sci. USA. 2017, 114, 6685–6688.

6.       Safaei, T.S.; Mepham, A.; Zheng X.L.; Pang, Y.J.; Dinh, C.T.; Liu, M.; Sinton, D.; Kelley, S.O.; Sargent, E.H. High-Density nanosharp microstructures enable efficient CO2 electroreduction. Nano Lett., 2016, 16, 7224–7228.

7.       Varela, A.S.; Kroschel, M.; Reier, T.; Strasser, P. Controlling the selectivity of CO2 electroreduction on copper: The effect of the electrolyte concentration and the importance of the local pH. Catal. Today 2016, 260, 8-13.

Round  2

Reviewer 3 Report

Authors replied to the reviewers' comments and questions well and revised the manuscript according to them, but not the conclusions yet. They concluded the copper nanowhiskers exhibited best CO2RR activity and favored C2H4 generation by suppressing CO production. However, in Figure 8, micro cluster copper synthesized under 20 mL/min flow rates suppressed more CO production and seems to show a better activity, but less C2/C1 selectivity than 40 mL. Further discussion and data interpretation are required for the conclusions, especially two structures vs. activity vs. selectivity.

Author Response

Point 1: Authors replied to the reviewers' comments and questions well and revised the manuscript according to them, but not the conclusions yet. They concluded the copper nanowhiskers exhibited best CO2RR activity and favored C2H4 generation by suppressing CO production. However, in Figure 8, micro cluster copper synthesized under 20 mL/min flow rates suppressed more CO production and seems to show a better activity, but less C2/C1 selectivity than 40 mL. Further discussion and data interpretation are required for the conclusions, especially two structures vs. activity vs. selectivity.

Response 1: Thanks for your remind and advice, we have rewritten the conclusion and added more discussion about structure vs. activity vs. selectivity. As indicated in the conclusion part, “The reduction product was mainly CO at lower overpotential, and high-valued ethylene became the main reduction product with increasing the cathodic potential and the synthesis of high curvature morphology. Morphologies synthesized under 20 mL/min and 40 mL/min flow rate showed relative high CO2RR activity, that is, rough surface with microclusters morphology exhibited favored methane and ethylene selectivity with C2/C1 ratio about 1, and nanowhiskers morphology exhibited favored ethylene selectivity (around 10% faradaic efficiency) with C2/C1 ratio of 1.8. According to previous mechanistic studies, the reaction pathways of CH4 and C2H4 differ in the bound CO* intermediates [37-40]. Both morphologies synthesized under 20 mL/min and 40 mL/min flow rate stabilized the bound CO* and prevented its desorption, which resulted in not only high reaction activity but also the further reaction pathways by hydrogenation of CO* or dimerization of two bound CO*. For rough Cu with microclusters morphology, aforementioned two pathways were coexisting and equi-favored, thus FE% for methane and ethylene were both around 5%. For nanowhiskers morphology, CO* hydrogenation to COH* could be suppressed by the existence of copper oxide and high local pH [38, 39], thus shifted the reaction towards C2H4 by stabilizing the OCCOH* intermediates (CO* + CO* + H+ + e- → OCCOH* → C2H4).”

Reviewer 4 Report

Satisfactory changes have been incorporated in manuscript  The manuscript can be considered for publication in Materials.

Author Response

Point 1: Satisfactory changes have been incorporated in manuscript. The manuscript can be considered for publication in Materials.

Response 1: Thanks for your effort and support on our work. We appreciate it.